# BNT162b2 Elicited an Efficient Cell-Mediated Response against SARS-CoV-2 in Kidney Transplant Recipients and Common Variable Immunodeficiency Patients

**DOI:** 10.3390/v15081659

**Published:** 2023-07-30

**Authors:** Evelina La Civita, Carla Zannella, Stefano Brusa, Paolo Romano, Elisa Schettino, Fabrizio Salemi, Rosa Carrano, Luca Gentile, Alessandra Punziano, Gianluca Lagnese, Giuseppe Spadaro, Gianluigi Franci, Massimiliano Galdiero, Daniela Terracciano, Giuseppe Portella, Stefania Loffredo

**Affiliations:** 1Department of Translational Medical Sciences, University of Naples “Federico II”, 80131 Naples, Italy; eva.lacivita@gmail.com (E.L.C.); stefano.brusa3@hotmail.it (S.B.); alessandra.punziano@unina.it (A.P.); g.lagnese93@gmail.com (G.L.); spadaro@unina.it (G.S.); daniela.terracciano@unina.it (D.T.); stefania.loffredo2@unina.it (S.L.); 2Department of Experimental Medicine, University of Campania “Luigi Vanvitelli”, 80138 Naples, Italy; carla.zannella@unicampania.it (C.Z.); massimiliano.galdiero@unicampania.it (M.G.); 3UOC of Virology and Microbiology, University Hospital of Campania “Luigi Vanvitelli”, 80138 Naples, Italy; 4Department of Public Health, Section of Nephrology, University of Naples “Federico II”, 80131 Naples, Italy; paolo.romano.92@alice.it (P.R.); elisa.schettino94@gmail.com (E.S.); fabriziosalemi@yahoo.it (F.S.); rosa.carrano@alice.it (R.C.); 5Integrated Department of Laboratory and Trasfusion Medicine, University of Naples “Federico II”, 80131 Naples, Italy; luca.gentile@unina.it; 6Center for Basic and Clinical Immunology Research (CISI), University of Naples “Federico II”, 80131 Naples, Italy; 7Department of Medicine, Surgery and Dentistry “ScholaMedicaSalernitana”, University of Salerno, 84081 Baronissi, Italy; gfranci@unisa.it; 8Clinical Pathology and Microbiology Unit, San Giovanni di Dio e Ruggi D’Aragona University Hospital, 84125 Salerno, Italy; 9Institute of Experimental Endocrinology and Oncology (IEOS), National Research Council, 80131 Naples, Italy

**Keywords:** mRNA vaccine, immunodeficiency, humoral immunity, cell-mediated immunity

## Abstract

SARS-CoV-2 vaccination is the standard of care for the prevention of COVID-19 disease. Although vaccination triggers both humoral and cellular immune response, COVID-19 vaccination efficacy is currently evaluated by measuring antibodies only, whereas adaptative cellular immunity is unexplored. Our aim is to test humoral and cell-mediated response after three doses of BNT162b vaccine in two cohorts of fragile patients: Common Variable Immunodeficiency (CVID) patients and Kidney Transplant Recipients (KTR) patients compared to healthy donors. We enrolled 10 healthy controls (HCs), 19 CVID patients and 17 KTR patients. HC BNT162b third dose had successfully mounted humoral immune response. A positive correlation between Anti-Spike Trimeric IgG concentration and neutralizing antibody titer was also observed. CVID and KTR groups showed a lower humoral immune response compared to HCs. IFN-γ release induced by epitopes of the Spike protein in stimulated CD4^+^ and CD8^+^ T cells was similar among vaccinated HC, CVID and KTR. Patients vaccinated and infected showed a more efficient humoral and cell-mediated response compared to only vaccinated patients. In conclusion, CVID and KTR patients had an efficient cell-mediated but not humoral response to SARS-CoV-2 vaccine, suggesting that the evaluation of T cell responses could be a more sensitive marker of immunization in these subjects.

## 1. Introduction

SARS-CoV-2 (Severe Acute Respiratory Syndrome Coronavirus 2) is the causative agent of the COVID-19 endemic. SARS-CoV-2 infected more than 625 million individuals causing over 6.5 million deaths worldwide up to October 2022 (Organization WH. WHO coronavirus (COV-19) dashboard (2022). Available at: https://covid19.who.int/ (accessed on 20 October 2022)).

Immunocompromised patients, such as solid organ transplant recipients and patients with weakened immune system are at increased risk of severe disease and death in case of infection [1].

Due to the harshly affected immune response to infection and immunization, Common Variable Immunodeficiency patients (CVID) and Kidney Transplant Recipients (KTR) patients represent a potential at-risk group in the current COVID-19 pandemic [2,3]. CVID is one of the most frequently diagnosed primary immunodeficiencies, found in about 1 in 25,000 persons, characterized by low levels of immunoglobulins (Ig) (IgG, IgA and/or IgM) [4]. The exact pathogenesis of CVID is still unclear, but the alteration in B cells maturation and development is a common feature of the disease. Although the hallmark of CVID is represented by frequent and severe bacterial infections, up to 50% of patients develop additional non-infectious complications including autoimmune manifestations, lymphoproliferation, enteropathy and granulomatous diseases [4]. The milestone of the treatment of CVID is represented by Immunoglobulin Replacement Therapy (IGRT), whose introduction has considerably reduced the frequency of infections, improving disease clinical course and survival [5]. Even if humoral response to vaccines is compromised, immunization with recombinant or inactivated vaccines is safe and strongly recommended [6].

Kidney transplant is a surgery done to replace a diseased kidney with a healthy kidney from a donor. The kidney may come from a deceased organ donor or from a living donor. A person getting a transplant most often gets just one kidney. In rare situations, he or she may get two kidneys from a deceased donor. The diseased kidneys are usually left in place. Conversely to CVID patients that have inborn errors of immunity, the immunosuppression for KTR is induced by combination treatment with drugs that lower the body’s ability to reject a transplanted organ [7].

SARS-CoV-2 vaccination is the standard for the prevention of COVID-19, with a positive impact in countries in which vaccination has been promoted [8].

However, waning of neutralizing antibodies after two doses of vaccine was observed in healthy and immunocompromised individuals [9]. Therefore, since the emergence of variants of concern (VOCs), European Medicines Agency (EMA) recommended a booster dose of the COVID-19 vaccines Comirnaty (BioNTech/Pfizer) and Spikevax (Moderna) for patients with severely weakened immune system and booster doses for subjects with normal immune system to ensure a long-lasting response.

Vaccines elicit long-term antigen-specific antibody responses by plasma cells, cell-mediated immunity response, and persistent memory development by T cells and B cells [10].

Though vaccination triggers both humoral and cellular immune response, COVID-19 vaccination efficacy is commonly evaluated by measuring only anti S antibodies concentration. However, a lower serological response to vaccination is a well-known problem in immunocompromised solid organ recipients due to the chronic immunosuppressive state [11]. Conversely, adaptative cellular immunity is poorly explored and considered in these pathological conditions. 

The aim of this study is to evaluate the humoral and cellular immune response after three doses of BNT162b vaccine in two cohort of fragile patients: CVID and KTR compared to healthy donors.

## 2. Materials and Methods

### 2.1. Study Population 

In this prospective study we included 19 CVID patients and 17 Kidney Transplant Recipient (KTR) patients enrolled from the Primary Immunodeficiency (PID) center and the Nephrology Unit of the University “Federico II” of Naples. In addition, 10 healthy volunteers were enrolled as controls.

CVID patients were diagnosed according to European Society for Immunodeficiencies (ESID) criteria (https://esid.org (accessed on 20 October 2022)) followed by the PID center at University “Federico II” of Naples. The CVID cohort was composed of 10 male and 9 female patients, with a median age of 41 years (IQR: 32–55). Patients were classified according to Chapel et al. into 2 main clinical phenotypes: 47% “infections only” and 53% “complicated”. There were a total of six patients with a story of autoimmune manifestation, five of them with immune thrombocytopenia (ITP) and one with vitiligo [4]. At the time of the study, all patients were in treatment with immunoglobulin replacement therapy at the dosage of 400 mg/kg/month; in 4 of them (21.1%), therapy was administered intravenous (IVIG) every 3–4 weeks, and through subcutaneous route (SCIG) in 15 patients (78.9%) every 10 days. 

All KTR patients received a kidney from adult brain-dead donors. The KTR cohort was composed of 13 male and 4 female patients with a median age of 50.7 (IQR: 42–62). The median transplant time to the booster dose was 11.7 years (IQR: 2–15). At the time of the study, 10 patients were in treatment with Tacrolimus, 4 with Ciclosporin, 1 with Sirolimus, 1 with Everolimus plus Tacrolimus and 1 Everolimus plus Ciclosporin every day. 

From March 2021, according to Italian vaccination program (https://www.salute.gov.it (accessed on 20 October 2022)), all our fragile patients underwent SARS-CoV2 immunization by BNT162b2 (Pfizer-BioNtech^®^, New York, NY, USA) and/or mRNA-1273 (Moderna^®^, Cambridge, MA, USA), in 2 doses followed by a booster dose between November 2021 and January 2022. 

Blood samples were collected after 21 days (±7 days, range from 14 to 28 days) the booster dose.

All patients and healthy donors signed a written consent form, and the study was approved by the local Ethical Committee (140/20/ESCOVID19). The regarding human material was managed using anonymous numerical codes, and all samples were handled in compliance with the Helsinki Declaration (http://www.wma.net/en/30publications/10policies/b3/ (accessed on 27 March 2021)).

Inclusion criteria for this study were compliance to vaccination program and expression of written informed consent. Exclusion criteria were presence of any condition that, in the opinion of the investigators, could interfere with the completion of the study.

CVID is often associated with comorbidities, and 31.6% of enrolled CVID patients presented comorbidities. In particular, 6 of 19 patients were affected by autoimmune manifestations: 5 by immune thrombocytopenia and 1 by vitiligo. Optimal management requires monitoring this complication when present. All KTR patients presented associated disorders such as hypertension (15 patients), ischemic heart disease (1 patient) and diabetes (2 patients). Patients’ comorbidities are reported in Table 1. 

### 2.2. Samples Collection

Whole blood samples were collected at admission using BD vacutainer (Becton Dickinson, Oxfordshire, UK) blood collection tubes containing EDTA and then immediately analyzed for lymphocyte count (Appendix A). Serum samples were separated from blood cells centrifugating samples at 3000 RCF 10′ after the collection in vacutainer tubes with no additives and immediately stored at −80 °C. In order to assess the lymphocytes response to spike protein, blood was collected in QuantiFERON Starter Set Blood Collection Tubes comprising Ag1 tubes that contain epitopes from the S1 subunit of the spike protein stimulating CD4^+^ T cell and an Ag2 tube with epitopes derived from the S1 and S2 subunits of the spike protein stimulating both CD4^+^ and CD8^+^ T cell. Nil and mitogen were used for negative and positive control, respectively. After the incubation at 37 °C for 24 h, plasma was harvested from the top layer of the tube by gently pipetting and immediately stored at −80 °C until the ELISA test was performed [12].

### 2.3. Determination of Released Interferon-γ in Plasma

Plasma from the stimulated samples was used for the measurement of Interferon-γ (IFN-γ). IFN-γ was evaluated using the QuantiFERON SARS-CoV-2—QIAGEN assay (Qiagen, Hilden, Germany) in accordance with the manufacturer’s instructions. Following ELISA, quantitative results (IFN-γ concentration in IU/mL) were generated by subtracting the ‘Nil’ values from samples and interpolating values with the standard curve.

### 2.4. Humoral Antibody Response

Humoral responses were measured with DiaSorinTrimericS IgG (DiaSorin, Saluggia, Italy) for the detection of IgG against the SARSCoV-2 Spike (S) protein receptor binding domain and Elecsys Anti-SARS-CoV-2 immunoassay (Roche Diagnostics, France) for the quantification of total antibodies (immunoglobulin (Ig) G, IgA, IgM) against the SARS-CoV-2 Nucleocapsid (N) antigen.

A cutoff of ≥33.8 BAU/mL was retained as positive for IgG anti-S, suggesting an effective vaccine-related immune response. Sera with anti-N antibodies concentration ≥1.0 was considered reactive indicating a virus contact [13].

### 2.5. SARS-CoV-2 Neutralization Assay (MNA)

Serum samples were maintained at −80 °C, thawed at the moment of the experiment, and serially diluted (1:10; 1:40; 1:160; 1:640) in Dulbecco’s Modified Eagle’s Medium (DMEM) supplemented with 1% Fetal Bovine Serum (FBS). Sera were then mixed with 100 TCID50 of SARS-CoV-2 and transferred in 96-well plates containing 5 × 10^5^ cells/mL Vero E6 cells. Plates were incubated at 37 °C for 72 h prior evaluation of cytopathic effect via microscope; then, they were fixed, stained with crystal violet solution and read at a spectrophotometer. The neutralization percentage was obtained by setting the mean OD595 of the serum control equal to 100%. Neutralization titers of serum samples were calculated by the highest serum dilution protecting 50% of the infected wells.

### 2.6. ELISA Assay

Plasma from the stimulated samples was used for the measurement of IL-6, CXCL8, TNF-α. IL-10 concentrations using commercially available ELISA Kits (R&D System, Minneapolis, MN, USA) according to the manufacturer’s instructions. Plasma concentrations of these mediators from CVID and KTR patients and HC were expressed as pg/mL. The sensitivity of the assay was 0.7 pg/mL for IL-6, 7.5 pg/mL for CXCL8, 7.21 pg/mL for TNF-α, 5.22 pg/mL for IL-10. The assay range was 3.1–300 pg/mL for the IL-6, and 31.2–2000 for the CXCL8, 10.9–700 pg/mL for TNF-α and 15.6–1000 pg/mL for IL-10. 

### 2.7. Statistical Analysis

Data were analyzed with the GraphPad Prism 5 software package. Data were tested for normality using the D’Agostino–Pearson normality test. If normality was not rejected at 0.05 significance level, we used parametric tests. Otherwise, for not-normally distributed data we used nonparametric tests. Statistical analysis was performed by unpaired two-tailed t-test or two-tailed Mann–Whitney test, as indicated in figure legends. Correlations between two variables were assessed by Spearman’s correlation analysis and reported as coefficient of correlation (*r*). A *p* value of ≤0.05 was considered statistically significant. Serum levels of Anti-Spike Trimeric IgG serum concentrations, Neutralization titers, Anti-SARS-CoV-2 Nucleocapsid (N) IgG, IFN-γ, IL-6, CXCL8, TNF-α. IL-10 V are shown as the median (horizontal black line), the 25th and 75th percentiles (boxes) and the 5th and 95th percentiles (whiskers) of study population.

## 3. Results

### 3.1. Serum Levels of IgG Directed against the (Trimeric) SARS-CoV-2 Spike-Proteinin CVID and KTR Patients after the Third Dose of mRNA Vaccine

In our cohort, the average lymphocytes value was 1628.23 cells/µL, the average IgG trough level (IgG-TL) was 601.75 mg/dL in IVIG patients and 673.692 mg/dL in SCIG patients; the average of lymphocytes of KTR cohort was 1574.70 cells/µL (Appendix A). 

First, to evaluate natural infection in our cohort of patients, we measured anti-N antibodies concentration. We found 5 CVID patients and 3 KTR patients with anti-N antibodies concentration higher than the cut-off (Table 1). Conversely, we did not find anti-N antibodies in HC. 

Next, we measured Anti-Spike Trimeric IgG concentration 21 days (±7 days, range from 14 to 28 days) after the third vaccination in 14 CVID anti-N negative patients, 14 KTR anti-N negative patients and 10 healthy controls. All study populations were not infected and had positive serum levels of Anti-Spike Trimeric (>33.8 BAU/mL). Figure 1A shows that the concentration of Anti-Spike Trimeric IgG was significantly lower in CVID and KTR patients compared to healthy controls (Healthy: 11,750 ± 12,319 BAU/mL ± SD; CVID: 137 ± 2161 BAU/mL± SD and KTR: 1295 ± 1109 BAU/mL± SD). No relevant differences in age and sex were found at this time point in IgG levels.

It is known that the development of neutralizing antibodies is correlated to virus protection, so we evaluated the neutralization titer of Anti-Spike Trimeric IgG. The titer was higher in controls than CVID patients (Figure 1B), and no significant differences were observed between HC and KTR, and CVID and KTR. In addition, we observed in a positive correlation between Anti-Spike Trimeric IgG concentration and neutralizing antibody titer (Figure 1C). 

### 3.2. QFN SARS-CoV-2 Assay in Vaccinated, but Not Infected Subjects

Next, we evaluated cell-mediated immunity at 21 days (±7 days, range from 14 to 28 days) after the third vaccination. To this aim, aAg1, which contains epitopes from the S1 subunit of the spike protein stimulating CD4^+^T cell and an Ag2 with epitopes derived from the S1 and S2 subunits of the spike protein stimulating both CD4^+^ and CD8^+^ T cell were used. An elevated IFN-γ response in Ag1 and Ag2 tubes was observed in most vaccinated subjects following third vaccine dose (Figure 2A,B). Interestingly, contrary to humoral response, the production of IFN-γ induced by Ag1 (Figure 2A) in healthy subjects was similar to that of CVID and KTR patients. Similar results were observed in the production of IFN-γ induced by Ag2 (Figure 2B). In particular, IFN-γ production is similar between HC and CVID patients while there is a slightly but not significant decreased od IFN-γ production induced by Ag2 by KTR. As expected, IFN-γ induced by Ag1 was positively correlated with that induced by Ag2 (Figure 2C) in the study population. Moreover, there was a positive correlation between humoral and cell-mediated immunity. In fact, IgG levels enhanced as IFN-γ induced by Ag1 and Ag2 increased (Figure 2D,E). 

### 3.3. Cytokine and Chemokine Concentration upon Ag1 and Ag2 Stimulation in Plasma of Vaccinated, but Not Infected Subjects

In addition to IFN-γ release, we evaluated the release of immunomodulator factors in our study population upon stimulation with Ag1 and Ag2. Figure 3 shows that the production of proinflammatory mediators such as IL-6 (panel A–B), CXCL8 (panel C–D) and TNF-α (panel E–F) induced by Ag1 and Ag2 was similar in, CVID, KTR patients and healthy controls. The release of anti-inflammatory cytokine IL-10 (panel G–H) did not differ in the three populations studied. 

### 3.4. Humoral and Cell-Mediated Response to SARS-CoV-2 in Hybrid Immunized CVID and KTR Patients

The immune response to SARS-CoV-2 is induced through natural infection or vaccination. In a final series of experiments, we evaluated humoral and cell-mediated SARS-CoV-2 immunity in CVID and KTR patients only vaccinated or vaccinated and infected (hybrid immunity). 

We divided the CVID and KTR patients according to the absence and presence of antibodies anti-nucleocapsid-IgG (N), elicited only in subjects with natural immunization. Figure 4A shows that CVID and KTR Anti N^+^ patients had higher concentration of Anti-Spike Trimeric IgG compared to Anti N-patients [CVID Anti-N^−^ 1010 ± 2162 BAU/mL (Mean ± SD) vs. CVID Anti-N^+^ 1611 ± 1185 BAU/mL; KTR Anti-N^−^ 1275 ± 1109 BAU/mL vs. KTR Anti-N^+^ 7313 ± 1204 BAU/mL]. The same result was obtained testing IFN-γ release. CVID and KTR patients with hybrid immunity had higher amounts of IFN-γ induced by Ag1 compared to only vaccinated patients. (Figure 4B) [CVID Anti-N^−^ 0.44 ± 0.42 IU/mL (Mean ± SD) vs. CVID Anti-N^+^ 0.788 ± 0.866 IU/mL; KTR Anti-N^−^ 0.29 ± 0.40 IU/mL vs. KTR Anti-N^+^ 1.81 ± 1.11 IU/mL] and Ag2 (Figure 4C) [CVID Anti-N^−^ 1.13 ± 1.64 IU/mL (Mean ± SD) vs. CVID Anti-N^+^ 2.32 ± 2.31 IU/mL; KTR Anti-N^−^ 0.38 ± 0.57 IU/mL vs. KTR Anti-N^+^ 2.2 ± 1.34 IU/mL].

## 4. Discussion

Here, we demonstrated that immunocompromised patients, CVID and KTR, after the third dose of mRNA COVID-19 vaccine had lower levels of serum IgG directed against the (trimeric) SARS-CoV-2 Spike-protein compared to healthy controls. A positive correlation between Anti-Spike Trimeric IgG concentration and neutralizing antibody titer was observed in our population. Conversely, the release of IFN-γ induced by epitopes derived from the S1 and S2 subunits of the Spike protein in stimulated CD4^+^ and CD8^+^ T cells was similar between vaccinated controls, CVID and KTR patients.

Moreover, according to previous studies, our data confirm that increased protection with hybrid immunity versus vaccine immunity in healthy and immunocompromised patients [14,15].

Vaccines play a vital role in protecting the host against infectious disease. Vaccine-mediated immunity is multifactorial, and the best protection is likely caused by the combination of humoral and cell-mediated immune responses [10]. The development of vaccines to prevent SARS-CoV-2 infection has mainly relied on the induction of neutralizing antibodies (nAb) to the Spike protein of SARS-CoV-2, but T cell immune response can contribute to protection as well [16]. 

Worthy of particular attention is the cell immune response to SARS-CoV-2 vaccine of CVID and KTR patients. Boyarsky et al. previously showed that after 20 days from the first dose of SARS-CoV-2 messenger RNA vaccine, antibodies were found in only 17% of a group of solid organ transplant recipients [17] and after 29 days from the second dose only in 54% [18]. Thus, about 50% of transplant recipients did not develop antibodies after two doses of mRNA-based vaccines. Other authors confirmed that transplant patients showed an insufficient humoral response to two doses of vaccination [19,20]. 

Nielsen et al. showed that multiple doses of mRNA SARS-CoV-2 vaccine are required in CVID patients to improve the antibody response and increase seroconversion in non-responders [21]. More recently, it has been demonstrated that, as expected, CVID patients did not produce strong humoral immune response after vaccination [22]. According to other authors, our study showed that third dose of BNT162b2 against SARS-CoV-2 infection induced a massive production of nAb IgG in healthy control. As expected, the Anti-SARS-CoV-2IgG production was significantly lower in CVID patients that have an inborn Ab deficiency. The same results were observed in KTR patients undergoing constant immunosuppressive maintenance therapy. Interestingly, neutralizing power of IgG was lower in CVID patients compared to healthy control and KTR. Unlike artificial immunized subjects (only vaccinated), infected and then vaccinated (hybrid immunized) healthy subjects had a more efficient humoral immune response [23]. Similarly, IgG production in vaccinated CVID and KTR patients was lower than hybrid immunized patients. Regarding cellular immune response, there is growing evidence showing that cellular immune response is much more preserved in solid organ transplant recipients than humoral immunity [24].

Some authors showed a severely impaired humoral response in immunodepressed and immunosuppressed patients even after several exposures of vaccine, but with a strong T-cell response. Evaluating both humoral and cellular responses in vaccinated patients with immune system failure appeared to be a good strategy to better assess individual immune protection [25,26,27,28].

Here, we investigated cell-mediated immune response following SARS-CoV-2 natural infection and/or COVID-19 vaccination using QuantiFERON SARS-CoV-2 assay. We found that the release of IFN-γ induced by epitopes derived from the S1 and S2 subunits of the Spike protein from stimulated CD4^+^ and CD8^+^ T cells of both CVID and KTR vaccinated patients was similar compared to vaccinated healthy subjects. The same results were observed when we tested other immunomodulating factor release such as IL-6, CXCL8, TNF-α, and IL-10. As for humoral response, the hybrid immunized patients had a cell-mediated response, as observed in IFN-γ production, which was harder than vaccinated patients. In our study population, despite the humoral response of CVID and KTR, patients who took a third dose of mRNA vaccine were impaired in respect to healthy controls, meaning the cell-mediated immunity worked effectively. Thus, in our study population, lower seroconversion rates evaluating the cellular immune response may be advantageous and more informative. However, the currently available assays are manual and time consuming; more automated and easy-to-perform assays for the measurement of Interferon-γ would be beneficial for a better evolution of immunocompromised patients. 

It is worth noting that, vaccine-induced cellular responses showed a stronger cross-protection against viral variants of concerns (VOCs) compared to humoral immunity [29].

The effectiveness of cell-mediated immunity after SARS-CoV-2 natural infection and/or COVID-19 vaccination would explain why, despite a high infection and reinfection rate and a high COVID-19-related mortality rate, the pandemic did not modify the annual mortality rate for any cause in a population of adult patients with CVID [30]. 

One strength of our study is the knowledge of the baseline patient characteristics, which may provide insights on factors associated with poor vaccine response. 

Nevertheless, the study has some limitations. Firstly, our results are related to the early post-vaccine period. Thus, we cannot evaluate how this immune response will protect the immunocompromised patients against SARS-CoV-2 infection and future variants. Secondly, our data were not enough to assess whether a fourth booster dose produces harder protection against SARS-CoV-2 infection, since we could not evaluate the immune response after the first dose of vaccine.

Thirdly, our study population was small and further studies on larger population are needed to better clarify the scenario.

In conclusion, our study suggests that assessing cellular immunity for SARS-COV-2 in addition to humoral immunity could be useful, considering that cellular immunity plays a pivotal role against the virus and likely its variants. Subjects with weakened immune response have no correlation between humoral and cellular immunity, suggesting that the evaluation of T cell responses could be a more sensitive clinical marker of immunization in this setting of patients.

## Figures and Tables

**Figure 1 viruses-15-01659-f001:**
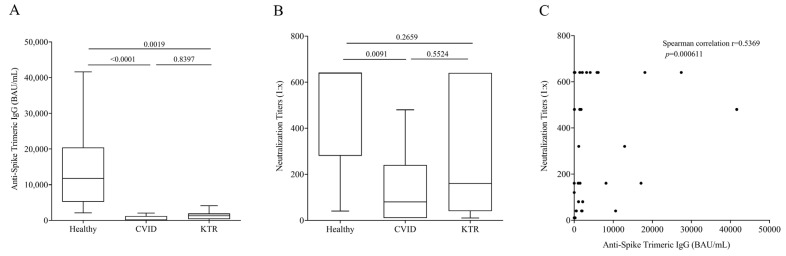
(**A**) Anti-Spike Trimeric IgG serum concentrations and (**B**) Neutralization titers 16–25 days after the third vaccination in Common Variable Immunodeficiency (CVID) patients, Kidney Transplant Recipients (KTR) and healthy donors. Data are shown as the median (horizontal black line), the 25th and 75th percentiles (boxes) and the 5th and 95th percentiles (whiskers) of 14 CVID, 14 KTR patients and 10 controls. (**C**) Correlation between Anti-Spike Trimeric IgG and Neutralization titers was assessed by Spearman’s correlation analysis and reported as coefficient of correlation. *p* < 0.05 was considered statistically significant.

**Figure 2 viruses-15-01659-f002:**
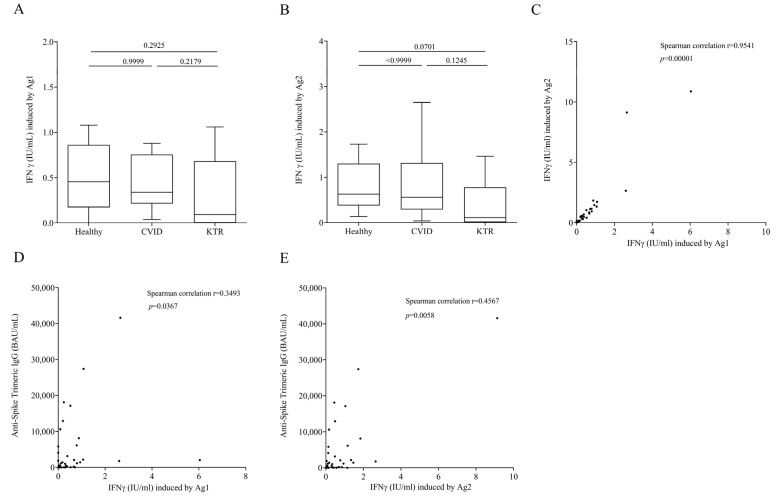
INF-γ plasma levels after exposure of lymphocytes to SARS-COV-2 Ag1 (**A**) and Ag2. (**B**) Peptides 16–25 days after the third vaccination and not infected in healthy controls, CVID and KTR. Data are shown as the median (horizontal black line), the 25th and 75th percentiles (boxes) and the 5th and 95th percentiles (whiskers) of 14 CVID, 14 KTR patients and 10 controls. (**C**) Correlations between two variables: INF-γ induced by Ag1 and INF-γ induced by Ag1, andINF-γ induced by Ag1 and Anti-Spike Trimeric IgG. (**D**) INF-γ induced by Ag2 and Anti-Spike Trimeric IgG. (**E**) Levels in study population were assessed by Spearman’s correlation analysis and reported as coefficient of correlation. *p* < 0.05 was considered statistically significant.

**Figure 3 viruses-15-01659-f003:**
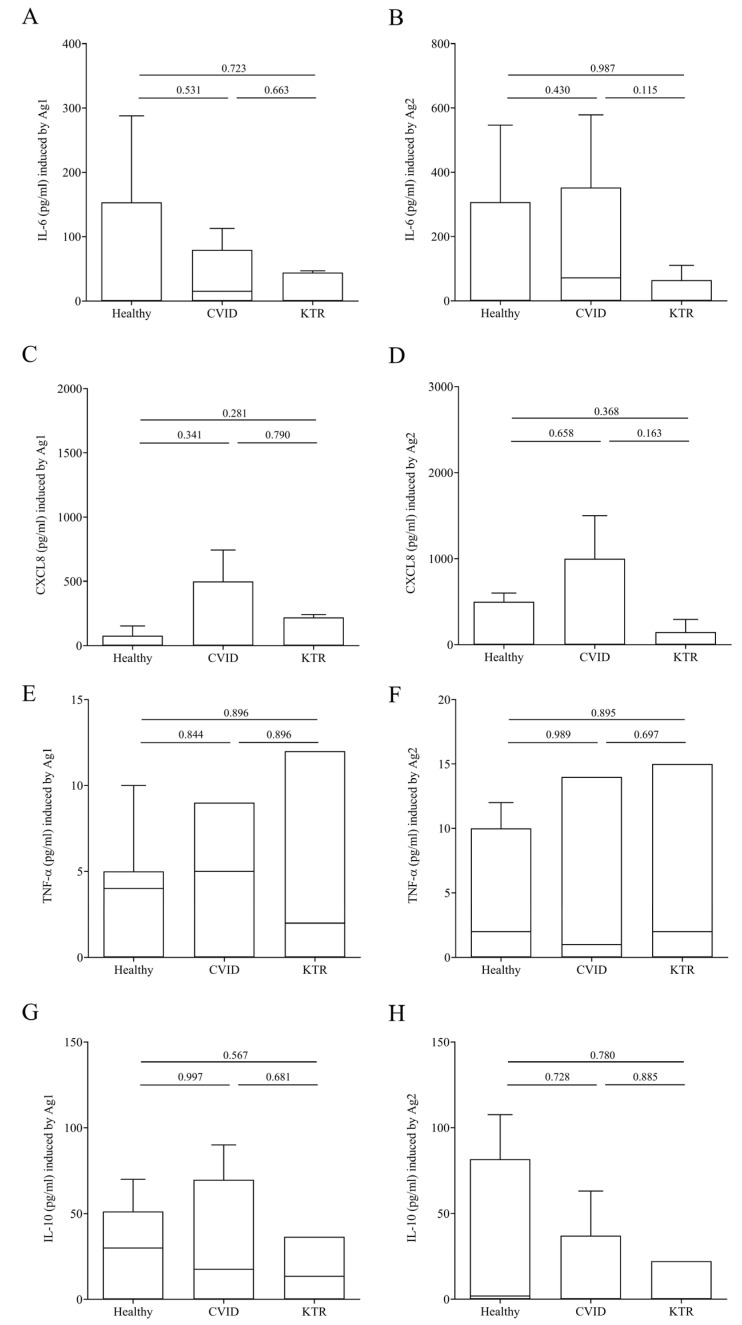
IL-6 (**A**,**B**), CXCL8 (**C**,**D**), TNF-α (**E**,**F**) and IL-10 (**G**,**H**) plasma levels after exposure of lymphocytes to SARS-COV-2 Ag1 (**A**,**C**,**E**,**G**) and Ag2 (**B**,**D**,**F**,**H**) peptides 16–25 days after the third vaccination in CVID, KTR and healthy controls. Data are shown as the median (horizontal black line), the 25th and 75th percentiles (boxes) and the 5th and 95th percentiles (whiskers) of 14 CVID, 14 KTR patients and 10 controls. *p* < 0.05 was considered statistically significant.

**Figure 4 viruses-15-01659-f004:**
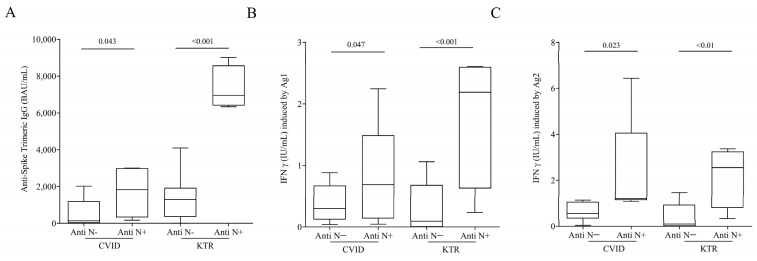
(**A**) Anti-Spike Trimeric IgG serum concentrations, INF-γ plasma levels after exposure of lymphocytes to SARS-COV-2 Ag1 (**B**) and Ag2 (**C**) peptides in infected and vaccinated CVID and in infected and vaccinated KTR patients with or without Anti-SARS-CoV-2 Nucleocapsid (N) IgG. Data are shown as the median (horizontal black line), the 25th and 75th percentiles (boxes) and the 5th and 95th percentiles (whiskers) of CVID andKTR patients. *p* < 0.05 was considered statistically significant.

**Table 1 viruses-15-01659-t001:** Characteristics of the study population.

Characteristics	Healthy (*n* = 10)	CVID (*n* = 19)	KTR (*n* = 17)
Age (y), median (range)	39 (27–69)	41 (23–74)	50.7 (24–77)
Gender Male (%)	4 (40%)	10 (52.6%)	13 (76.4%)
Gender Female (%)	6 (60%)	9 (47.3%)	4 (23.5%)
Caucasian race (%)	10 (100%)	19 (100%)	16 (94.1%)
Anti-N antibodies positive		5 (26.3%)	3 (17.6%)
Comorbidity (%)	0	6 (31.6%)	17 (100%)
Autoimmune manifestation		6	
Immune thrombocytopenia		5	
Vitiligo		1	
Hypertension			15
Ischemic heart disease			1
Diabetes			2

## Data Availability

Data and material supporting the reported results are available upon request.

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
