# Peer review of "BNT162b2 Elicited an Efficient Cell-Mediated Response against SARS-CoV-2 in Kidney Transplant Recipients and Common Variable Immunodeficiency Patients"

_viruses, 2023, doi:10.3390/v15081659_

Round 1
Reviewer 1 Report
Civita et al measured the humoral and cellular immune response induced by complete SARS-CoV-2 vaccination (2 doses for primary immunization + 1 booster dose) in people with CVID (n = 17?), KTRs (n = 17?) and healthy controls (n = 10). Humoral immune response was assessed by measuring IgG levels and testing neutralization capacity of participants antibodies. For cellular response IFNg as well as other cytokines and chemokines were measured.
Overall, non content related comment: There appears to be something of with formatting of this manuscript. At multiple instances words have been joined (e.g. amore instead of a more), sudden changes in font size have been discovered and so on. Besides that, the manuscript must be carefully revised – in some instances words are missing from sentences...
Minor comment regarding the following sentence in the methods section: >>After the incubation at 37°C for 24h, plasma was harvested from the top layer of the tube by gently pipetting and immediately stored at –80°C until the ELISA test was performed<< Tubes were centrifuged before, right?
Comments:
The provided patient characteristics are very sparse. For KTR for instance it is known that triple IS and shorter time after transplant are major risk factors for vaccine non-response.
According to the abstract 17 CVID and 17 KTR were included in the study. In the results section it is, however, described that anti spike ab were measured in 14 CVID and 14 KTR only.
The authors state NT titers were higher in KTR than in CVID patients but the two distributions do not significantly differ.
While there is a positive correlation between spike IgG and NT it is rather low (0.5) and may be mainly driven by the few dots with higher IgG levels (are these predominantly HC)?
The authors claim that elevated IFNg levels were observed in most individuals after vaccination. The current data only support detecting an elevation compared to the NIL tube as there is neither pre-vac data for this individuals nor IFNg levels for SARS-CoV-2 naïve (no infection no vaccination) people reported.
In sentence 209 the authors suggest that IFNg levels were similar between HC and KTR. The p-value provided in the boxplot could already be considered borderline significant.
Correlation between IFNg and IgG levels appears weak.
It appears as IL-6 and CXCL8 were not detectable in at least half of the healthy controls. Was the test system sensitive enough? Plots are missing p-values for group comparisons.
In the last part of the paper the authors stratify based on hybrid immunization status. Where do these SARS-CoV-2 experienced patients come from? In the first part of the results it is written >>All study population was not infected and had positive serum levels of Anti-Spike Trimeric (>33.8 BAU/ml)<<. Does this solely mean that none of the patients had an active SARS-CoV-2 infection during the study period, but that the study population comprised SARS-CoV-2 naïve and experienced people. If yes, this needs to be addressed since any reported immune response to the vaccine may not be fully attributable to the vaccine itself but also to previous infections.
The authors claim that a strength of their study is that baseline characteristics of the patients are known an that this would allow to identify risk factors. The baseline characteristics are however very sparse (despite assignment to the two risk groups) and the samples size will limit the ability to identify any risk factors.
-
Author Response
Reviewer 1
Civita et al measured the humoral and cellular immune response induced by complete SARS-CoV-2 vaccination (2 doses for primary immunization + 1 booster dose) in people with CVID (n = 17?), KTRs (n = 17?) and healthy controls (n = 10). Humoral immune response was assessed by measuring IgG levels and testing neutralization capacity of participants antibodies. For cellular response IFNg as well as other cytokines and chemokines were measured.
Overall, non content related comment: There appears to be something of with formatting of this manuscript. At multiple instances words have been joined (e.g. amore instead of a more), sudden changes in font size have been discovered and so on. Besides that, the manuscript must be carefully revised – in some instances words are missing from sentences...
Minor comment regarding the following sentence in the methods section: >>After the incubation at 37°C for 24h, plasma was harvested from the top layer of the tube by gently pipetting and immediately stored at –80°C until the ELISA test was performed<< Tubes were centrifuged before, right?
Comments:
The provided patient characteristics are very sparse. For KTR for instance it is known that triple IS and shorter time after transplant are major risk factors for vaccine non-response.
Thank you for this valuable suggestion. We modified table 1, adding other patient characteristics.
According to the abstract 17 CVID and 17 KTR were included in the study. In the results section it is, however, described that anti spike ab were measured in 14 CVID and 14 KTR only.
We thank the reviewer for this comment. We changed the abstract and the text with the right numbers.
The authors state NT titers were higher in KTR than in CVID patients, but the two distributions do not significantly differ.
We agree with the reviewer. We modified the text highlighting that we observed a difference, but this was not significant.
While there is a positive correlation between spike IgG and NT it is rather low (0.5) and may be mainly driven by the few dots with higher IgG levels (are these predominantly HC)?
We thank the reviewer for this observation. As you can see by the following data, the few dots with higher IgG levels are actually from CVID and KTR patients and not from HC.
The authors claim that elevated IFNg levels were observed in most individuals after vaccination. The current data only support detecting an elevation compared to the NIL tube as there is neither pre-vac data for this individual nor IFNg levels for SARS-CoV-2 naïve (no infection no vaccination) people reported.
We are grateful to the reviewer for this comment. We added this aspect in the limitations of the study showed in the discussion section.
In sentence 209 the authors suggest that IFNg levels were similar between HC and KTR. The p-value provided in the boxplot could already be considered borderline significant.
We greatly appreciate this suggestion. We modified the text highlighting that there was a trend towards significance.
Correlation between IFNg and IgG levels appears weak.
We thank the reviewer for this observation. The weak correlation may be due to the low number of patients. This is a limitation of our study stressed in the discussion section.
It appears as IL-6 and CXCL8 were not detectable in at least half of the healthy controls. Was the test system sensitive enough? Plots are missing p-values for group comparisons.
Thank you for this precious comment. According to your observations, we added test system sensitivity in method section and p-values in the plots.
In the last part of the paper the authors stratify based on hybrid immunization status. Where do these SARS-CoV-2 experienced patients come from? In the first part of the results it is written >>All study population was not infected and had positive serum levels of Anti-Spike Trimeric (>33.8 BAU/ml)<<. Does this solely mean that none of the patients had an active SARS-CoV-2 infection during the study period, but that the study population comprised SARS-CoV-2 naïve and experienced people. If yes, this needs to be addressed since any reported immune response to the vaccine may not be fully attributable to the vaccine itself but also to previous infections.
We thank the reviewer for this observation. We evaluated active SARS-CoV-2 infection by measuring anti-N antibodies and we added the results in the text.
The authors claim that a strength of their study is that baseline characteristics of the patients are known an that this would allow to identify risk factors. The baseline characteristics are however very sparse (despite assignment to the two risk groups) and the samples size will limit the ability to identify any risk factors.
We really appreciate this comment. We modified table 1, adding other patient information. We are aware that samples size limit our ability to identify risk factors and we only suggest the potential of a study population with known baseline characteristics.
Reviewer 2
The authors have evaluated the humoral and cellular response to COVID-19 vaccine in three groups of patients- healthy controls, renal transplant recipients and individuals with immunodeficiency. Humoral response was reduced in the two study groups in comparison with controls; on the contrary, no difference occurred in cellular response between the three groups.
According to the investigators, anti-Spike Trimeric IgG antibody levels were reduced in CVID and KTR patients compared to healthy controls (Healthy: 11750±12319 BAU/ml± SD; CVID: 137±2161 BAU/ml± SD and 185 KTR: 1295±1109 BAU/ml± SD).
I suggest to change table 1, it needs to include not only age, gender, and race but various comorbidities with more details. I recommend that the authors should include some references supporting the topic of protection against viruses conferred from vaccine (via cellular immunity). Protection against viruses conferred from cellular arm (in addition to humoral arm) is a neglected topic among current clinicians.
Thank you for these valuable suggestions. We modified table 1 and we added some references in discussion section accordingly.
|
|

Reviewer 2 Report
The authors have evaluated the humoral and cellular response to COVID-19 vaccine in three groups of patients- healthy controls, renal transplant recipients and individuals with immunodeficiency. Humoral response was reduced in the two study groups in comparison with controls; on the contrary, no difference occurred in cellular response between the three groups.
According to the investigators, anti-Spike Trimeric IgG antibody levels were reduced in CVID and KTR patients compared to healthy controls (Healthy: 11750±12319 BAU/ml± SD; CVID: 137±2161 BAU/ml± SD and 185 KTR: 1295±1109 BAU/ml± SD).
I suggest to change table 1, it needs to include not only age, gender, and race but various comorbidities with more details. I recommend that the authors should include some references supporting the topic of protection against viruses conferred from vaccine (via cellular immunity). Protection against viruses conferred from cellullar arm (in addition to humoral arm) is a neglected topic among current clinicians.
Author Response
Reviewer 1
Civita et al measured the humoral and cellular immune response induced by complete SARS-CoV-2 vaccination (2 doses for primary immunization + 1 booster dose) in people with CVID (n = 17?), KTRs (n = 17?) and healthy controls (n = 10). Humoral immune response was assessed by measuring IgG levels and testing neutralization capacity of participants antibodies. For cellular response IFNg as well as other cytokines and chemokines were measured.
Overall, non content related comment: There appears to be something of with formatting of this manuscript. At multiple instances words have been joined (e.g. amore instead of a more), sudden changes in font size have been discovered and so on. Besides that, the manuscript must be carefully revised – in some instances words are missing from sentences...
Minor comment regarding the following sentence in the methods section: >>After the incubation at 37°C for 24h, plasma was harvested from the top layer of the tube by gently pipetting and immediately stored at –80°C until the ELISA test was performed<< Tubes were centrifuged before, right?
Comments:
The provided patient characteristics are very sparse. For KTR for instance it is known that triple IS and shorter time after transplant are major risk factors for vaccine non-response.
Thank you for this valuable suggestion. We modified table 1, adding other patient characteristics.
According to the abstract 17 CVID and 17 KTR were included in the study. In the results section it is, however, described that anti spike ab were measured in 14 CVID and 14 KTR only.
We thank the reviewer for this comment. We changed the abstract and the text with the right numbers.
The authors state NT titers were higher in KTR than in CVID patients, but the two distributions do not significantly differ.
We agree with the reviewer. We modified the text highlighting that we observed a difference, but this was not significant.
While there is a positive correlation between spike IgG and NT it is rather low (0.5) and may be mainly driven by the few dots with higher IgG levels (are these predominantly HC)?
We thank the reviewer for this observation. As you can see by the following data, the few dots with higher IgG levels are actually from CVID and KTR patients and not from HC.
The authors claim that elevated IFNg levels were observed in most individuals after vaccination. The current data only support detecting an elevation compared to the NIL tube as there is neither pre-vac data for this individual nor IFNg levels for SARS-CoV-2 naïve (no infection no vaccination) people reported.
We are grateful to the reviewer for this comment. We added this aspect in the limitations of the study showed in the discussion section.
In sentence 209 the authors suggest that IFNg levels were similar between HC and KTR. The p-value provided in the boxplot could already be considered borderline significant.
We greatly appreciate this suggestion. We modified the text highlighting that there was a trend towards significance.
Correlation between IFNg and IgG levels appears weak.
We thank the reviewer for this observation. The weak correlation may be due to the low number of patients. This is a limitation of our study stressed in the discussion section.
It appears as IL-6 and CXCL8 were not detectable in at least half of the healthy controls. Was the test system sensitive enough? Plots are missing p-values for group comparisons.
Thank you for this precious comment. According to your observations, we added test system sensitivity in method section and p-values in the plots.
In the last part of the paper the authors stratify based on hybrid immunization status. Where do these SARS-CoV-2 experienced patients come from? In the first part of the results it is written >>All study population was not infected and had positive serum levels of Anti-Spike Trimeric (>33.8 BAU/ml)<<. Does this solely mean that none of the patients had an active SARS-CoV-2 infection during the study period, but that the study population comprised SARS-CoV-2 naïve and experienced people. If yes, this needs to be addressed since any reported immune response to the vaccine may not be fully attributable to the vaccine itself but also to previous infections.
We thank the reviewer for this observation. We evaluated active SARS-CoV-2 infection by measuring anti-N antibodies and we added the results in the text.
The authors claim that a strength of their study is that baseline characteristics of the patients are known an that this would allow to identify risk factors. The baseline characteristics are however very sparse (despite assignment to the two risk groups) and the samples size will limit the ability to identify any risk factors.
We really appreciate this comment. We modified table 1, adding other patient information. We are aware that samples size limit our ability to identify risk factors and we only suggest the potential of a study population with known baseline characteristics.
Reviewer 2
The authors have evaluated the humoral and cellular response to COVID-19 vaccine in three groups of patients- healthy controls, renal transplant recipients and individuals with immunodeficiency. Humoral response was reduced in the two study groups in comparison with controls; on the contrary, no difference occurred in cellular response between the three groups.
According to the investigators, anti-Spike Trimeric IgG antibody levels were reduced in CVID and KTR patients compared to healthy controls (Healthy: 11750±12319 BAU/ml± SD; CVID: 137±2161 BAU/ml± SD and 185 KTR: 1295±1109 BAU/ml± SD).
I suggest to change table 1, it needs to include not only age, gender, and race but various comorbidities with more details. I recommend that the authors should include some references supporting the topic of protection against viruses conferred from vaccine (via cellular immunity). Protection against viruses conferred from cellular arm (in addition to humoral arm) is a neglected topic among current clinicians.
Thank you for these valuable suggestions. We modified table 1 and we added some references in discussion section accordingly.
